# Similarity Analysis in Understanding Online News in Response to Public Health Crisis

**DOI:** 10.3390/ijerph192417049

**Published:** 2022-12-19

**Authors:** Sidemar Cezario, Thiago Marques, Rafael Pinto, Juciano Lacerda, Lyrene Silva, Thaisa Santos Lima, Orivaldo Santana, Anna Giselle Ribeiro, Agnaldo Cruz, Ana Claudia Araújo, Angélica Espinosa Miranda, Aedê Cadaxa, César Teixeira, Almudena Muñoz, Ricardo Valentim

**Affiliations:** 1Department of Informatics and Applied Mathematics, Federal University of Rio Grande do Norte, Natal 59078-900, Brazil; 2Laboratory for Technological Innovation in Health (LAIS), Federal University of Rio Grande do Norte, Natal 59010-090, Brazil; 3Information Systems Coordination, Federal Institute of Rio Grande do Norte, Natal 59015-300, Brazil; 4Department of Social Communication, Federal University of Rio Grande do Norte, Natal 59072-970, Brazil; 5Federal Senate, Brasília 70165-900, Brazil; 6School of Science and Technology, Federal University of Rio Grande do Norte, Natal 59078-970, Brazil; 7Ministry of Health, Brasília 70070-600, Brazil; 8Postgraduate Program in Infectious Diseases, Federal University of Espírito Santo, Vitória 29075-910, Brazil; 9Department of Informatics Engineering, Centre for Informatics and Systems of the University of Coimbra (CISUC), 3030-290 Coimbra, Portugal; 10Department of Communication Theories and Analysis, Complutense University of Madrid, 28040 Madrid, Spain; 11Department of Biomedical Engineering, Federal University of Rio Grande do Norte, Natal 59628-330, Brazil

**Keywords:** public health, health policy, data mining, text extraction, communicable disease, syphilis

## Abstract

Background: The “Syphilis No!” campaign the Brazilian Ministry of Health (MoH) launched between November 2018 and March 2019, brought forward the concept "Test, Treat and Cure" to remind the population of the importance of syphilis prevention. In this context, this study aims to analyze the similarity of syphilis online news to comprehend how public health communication interventions influence media coverage of the syphilis issue. Methods: This paper presented a computational approach to assess the effectiveness of communication actions on a public health problem. Data were collected between January 2015 and December 2019 and processed using the Hermes ecosystem, which utilizes text mining and machine learning algorithms to cluster similar content. Results: Hermes identified 1049 google-indexed web pages containing the term ’syphilis’ in Brazil. Of these, 619 were categorized as news stories. In total, 157 were grouped into clusters of at least two similar news items and a single cluster with 462 news classified as “single” for not featuring similar news items. From these, 19 clusters were identified in the pre-campaign period, 23 during the campaign, and 115 in the post-campaign. Conclusions: The findings presented in this study show that the volume of syphilis-related news reports has increased in recent years and gained popularity after the SNP started, having been boosted during the campaign and escalating even after its completion.

## 1. Introduction

In Brazil, reported cases of syphilis rose dramatically during 2010–2018 (Figure 1), leading the Brazilian Ministry of Health (MoH) to declare an epidemic of syphilis in 2016. Over that period, according to MoH data, the case detection rate of acquired syphilis increased from 34.1 cases per 100,000 population in 2015 to 75.8 in 2018 (rate of growth = 122.29%). The case detection rate of syphilis in pregnancy (SIP) jumped from 10 cases per 1000 live births in 2015 to 21.40 cases in 2018 (rate of growth = 96.33%). Ultimately, congenital syphilis (CS) cases went from 6.5 to 9 per 1000 live births, marking a 38.45% increase [1].

Then, in 2017, the MoH launched the “Syphilis No!” Project (SNP) [2,3,4]. With national coverage, the SNP was operationalized by dimensions in response to syphilis, in an inter-federative approach, within the Brazilian National Health System (SUS). For instance, the communication dimension aimed to disseminate the topic of syphilis nationwide [5]. Between November 2018 and March 2019, a national health communication campaign, whose main theme was “Test, Treat, and Cure”, was launched as one of the SNP activities. The campaign shed light on the problem of syphilis while helping to project the SNP on a national level [2,3].

Public health communication campaigns are of prime importance for the continuous care of population health [6,7]. Over the years, governmental sectors have been increasingly investing in public health campaigns to promote public awareness regarding disease risk, as well as to disseminate diagnostic tests and treatment methods [8]. What is more, as a strategy to increase the reach and dissemination of these sorts of campaigns, governmental entities sponsor the dissemination of news through posts on websites, social media, microblogs, and digital influencers, among others. The greater the dissemination of news items on the topic, the more likely the number of people reached by such information will be, thus improving the chances of early diagnosis, treatment, and cure of cases [9].

Besides sponsored news, there are those news items that people or institutions spontaneously convey [10,11]. This trend is not limited to the act of replicating information, for it underlies a public interest service/action—an aspect that calls for new ways of evaluating and measuring the impact of health communication campaigns.

Hence, evaluating news replication denotes not only the success of the agenda-setting process, which might have been induced but also the existence of an “intermedia agenda-setting” process by which one medium influences other news media [12,13]. Such a process acts on public opinion and organically reverberates in various other instances of communication.

The goals of this study are twofold: (1) to assess the massive number of syphilis-related news stories in Brazil during 2015–2019; and (2) to identify which news stories were the most disseminated or underpinned the production of further news, i.e., news items that have effectively prompted the “intermedia agenda-setting” effect. In order to achieve these goals, we used the Hermes ecosystem, a module for the analysis of news reports implemented in a digital health information ecosystem [2,3]. By using data mining and machine learning techniques, the Hermes collected and processed online syphilis-related news stories disseminated in Brazil from 2015 to 2019.

## 2. Materials and Methods

The effect of the syphilis online news clusters results renders substantial reflections on the intermedia agenda-setting process [12]. Figure 2 shows the theory that laid the groundwork for this study. In such a process [13], there is a latency period between the initial stimulus (A) and the correspondence as an effect that defines a newspaper of national or regional relevance, for instance, to set the agenda or be featured in a different media outlet (B). Subsequently, we have a period of increasing correlation between agendas (C), followed by a sustained period (C-D) and a decreasing period (D-E) [13].

Recent studies have demonstrated effectiveness in clustering large volumes of online news items [14,15,16]. Text clustering can be defined as keeping similar group documents together, marking the outliers texts outside the group based on similarity estimation between them [17]. Thus, after reviewing the current literature and comparing approaches, we conducted the following steps: (i) news collection; (ii) data cleaning and pre-processing; (iii) application of a clustering algorithm to similar texts; and (iv) analysis of the results.

### 2.1. Data Collection

We collected the web pages using the Hermes ecosystem, which is based on a multidimensional framework for temporal analysis of public health interventions [2]. This system collected 1049 Google-indexed web pages containing the term ‘syphilis’ in Brazil between January 2015 and December 2019, i.e., before, during, and after the “Syphilis No!” campaign. After collection, Hermes extracted the main content of these web pages, which communication experts subsequently analyzed in order to exclude non-news pages, including web portal search results (n = 71), non-textual content (files, videos, and podcasts; n = 124), scientific content pages (n = 185), unavailable pages or pages with members-only content (n = 47), and Frequently Asked Questions (FAQ) pages (n = 3). In this preliminary analysis, 619 web pages were classified as news.

### 2.2. Data Cleaning and Pre-Processing

To achieve more accurate results in clustering similar news, we conducted a four-stage text cleaning process: (i) stop-words removal (insignificant words, adverbs, and prepositions, for example); (ii) conversion of uppercase strings into lower cases; (iii) digit removal; and (iv) lemmatization, which consists in grouping the different inflected forms of syntactically different but semantically equivalent words. In this process, for example, terms describing the verbs “see” and “saw” are grouped under “see”.

As in the works by [17,18,19], our study used the Term Frequency-Inverse Document Frequency (TF-IDF) [20], a widespread method in the field of natural language processing (NLP). Thus, the news items were fragmented into a set of terms for the TF-IDF procedure to assign a value to the relevance of each term in the text. We ran several tests to define the best number of words to represent a single news report. Therefore, each news item was represented by a set of terms used by the clustering techniques.

### 2.3. Application of a Clustering Algorithm to Similar Texts

In this step, we sought to use an unsupervised clustering approach, namely Density-Based Spatial Clustering of Applications with Noise (DBSCAN) [21]. This algorithm is based on the premise of creating similar datasets and keeping distinct data in distinct clusters. DBSCAN requires two parameters for its operation. The first, ϵ, indicates the maximum distance for data to be considered from the same neighborhood. And the second, *s*, informs the minimum number of samples per cluster. This algorithm works as described below.
For each point in the dataset, it searches for neighbors that respect the distance ϵ, forming a cluster with at least s elements.For each cluster formed, the algorithm tries to expand it by adding more data points that do not yet belong to another cluster.For each point not in a cluster, if it respects the distance ϵ from some cluster, that point is added to the cluster; otherwise, the point is labeled as noise.

We set the parameter ϵ (distance between observations) to 0.5 (for better results) and the minimum number of instances to form a group to 2, as a replication consists of at least the original news and its reproduction.

### 2.4. Limitations

Our study has some limitations. One of them is related to the completeness of the news items collected, an external threat mitigated by choosing Google Search, one of the largest existing content indexers in the world. Nevertheless, we would not assume that Hermes can retrieve every existing online news through the Google Search API. In addition, Hermes used filters for (i) language, which reduces the results to documents written in Brazilian Portuguese, and (ii) geolocation, which limits the results to records originating from Brazil. That may comprise a barrier insofar as the geolocation parameter checks the domain and the geographical location of the Web server’s IP address.

## 3. Results

This section presents the results of our experiment on the collated online news using the DBSCAN algorithm. Table 1 breaks down the number of clusters as well as their sizes. We found 60 clusters formed by 157 news reports. What is more, 462 news items were classified as single, thus not belonging to any group.

To validate the solutions DBSCAN found, we manually read and analyzed the clusters. As a result, we identified that the news reports featured similar or identical excerpts in each group, which confirms such excerpts originated from a piece of news in common (see an example of a news report extracted from a cluster DBSCAN identified in the Appendix A). 

Table 2 depicts the distribution of news and clusters throughout three time periods: pre-campaign (1 January 2015 to 21 November 2018), during-campaign (22 November 2018 to 31 March 2019), and post-campaign (1 April 2019 to 31 December 2019). The pre-campaign period encompassed 164 news reports (26.49% of total news reports) and 18 clusters (11.46% of total clusters); the during-campaign period, 85 news reports (13.73% of total) and 24 clusters (15.29% of total), and lastly, the post-campaign period, 370 news reports (59.77% of total) and 115 clusters (73.25% of total).

Figure 3 provides the results of the replications observed, where each dot indicates the publication of a news item and the line represents the time interval between similar news. Each color represents a different cluster, and the dashed vertical lines mark off the campaign periods.

Regarding the during-campaign period, we verified that the proportion of replications in this period was 15.29%, a modest growth compared with the pre-campaign period. Moreover, by evaluating the replications that occurred post-campaign, one will observe a 73.25% increase, which indicates the campaign actions encouraged an organic diffusion of syphilis-related information.

Another issue deserving of investigation is that of single news (not grouped into clusters), i.e., news items that do belong to any cluster. Considering the exact scenarios for such news, we found that 26.49% of news was produced before the campaign, 13.73% during the campaign, and 59.77%, post-campaign. When evaluated from a managerial perspective, these results demonstrate that single news items are highly significant since they address different facets of syphilis.

From this viewpoint, by using time series decomposition analysis, we processed the data and plotted it into a trend graph to verify the conversion point in the news trend. According to Figure 4, there was a steady publication tendency of syphilis-related news in Brazil from September 2015 to February 2018. However, after March 2018 (the first dashed vertical line), it is possible to observe a sloping line that accentuates during the campaign (November 2018 to March 2019) and remains increasing by the end of the observed period, with no fall. Before the SNP, syphilis campaigns were integrated into HIV or other sexually transmitted infections (STIs) campaigns, focusing on SIP or CS, without much investment in resources and different types of media [22]. Adopting a distinct approach, with greater resources investment and media diversity, the 2018-2019 campaign highlighted collective aspects without attributing blame to individuals, promoting more diverse representations of target audiences [23].

The graph in Figure 4 displays an inflection point in the trend change that started in May 2018, which is justified because that month was effectively the month of the beginning of SNP activities across Brazil, subsuming several seminars and inter-federative actions [3]. This drew the local media’s attention and mobilized communication efforts among state and municipal health secretaries in Brazil.

Considering this perspective, it is reasonable to infer that the pre-campaign period, with 164 news items, had a low level of replications (11.46%), thus representing an earlier and more stable time frame. Nationwide diffusion of the SNP communication campaign, starting at the end of the third quarter of 2018, is considered a new initial incentive in this analysis. In addition, the latency period of the setting of the syphilis issue in the intermedia agenda is characterized by the during-campaign period, accounting for 85 news items and 13.73% of replications. Thus, the time span we refer to as "post-campaign," with 370 news items and 73.25% of replications, is marked by the period of correspondence between the setting of the topic on the Brazilian digital media agendas, where the syphilis issue was set in numerous other news outlets during the period of sustained coverage over time.

## 4. Discussion

The results of the clustering of syphilis online news during the three periods of the SNP campaigns corroborate the understanding of the dynamics and enhancement of public health interventions in the field of communication. There is solid evidence that combining mass media with other communication strategies is highly effective [6].

Such an effect can be observed in our results as the rise in syphilis online news is still noticeable even after the campaign’s end. Therefore, using techniques and methods applied to digital health technologies based on machine learning, such as those used in this article, becomes essential for measuring the impacts of communication agendas in public health. This aspect is critical in contexts of public health crises, such as the syphilis epidemic in Brazil, due to the urgency in measuring the effectiveness of public health response communication actions.

Notably, 74.64% (n = 462) of the news analyzed by the algorithms developed for this study were not grouped into clusters. Not belonging to a cluster implies that the news item is “single”, i.e., it was not replicated. We investigated this phenomenon for all the news items in the corpus with this hallmark. In this context, we observed that 100% of these news stories were effectively material of an authorial nature produced by the journalistic institutions or organizations themselves, with specific and qualitative approaches to the different themes of the disease. That is an interesting scientific finding of our study, for it validates the technique applied in this research method. For the area of management, which is concerned with mass communication in public health, the existence of algorithms with this distinction capacity is relevant to evaluating the quality of journalistic texts on health care.

All 462 news items identified as “single” were news/special reports from national or regional media, in addition to editorials, news pieces published by universities, NGOs, and health organizations, or op-ed articles. On the website (digital address) in which single news stories were published (Table 3), we identified a nearly proportional distribution between national (n = 68) and regional (n = 86) corporate media and original materials posted by the MoH and state and municipal health secretariats (162) or health or societal organizations or educational institutions (146). Of note, the most significant amount of news stories in the national and regional press outnumbered publications by government and societal institutions.

As for the journalistic genre [24] (Table 4), the set of 462 single news primarily includes news items (n = 295), i.e., information based on recent events or updates about already well-known facts. Next, we have the opinion articles (n = 73) representing the singular voice of journalists, experts, or renowned figures who express their opinions about certain aspects of the syphilis problem. The news reports (n = 71) identified are texts that, in addition to presenting syphilis-related facts, provide a more in-depth editorial treatment and a plurality of voices and data, aiming to contextualize, situate, or present readers with perspectives on the infection. Informative materials were classified under the category "Other." These include podcasts, murals, reports, glossaries, and entries that present significant data on syphilis but lack the temporality and factual emergency that characterize news stories and reports.

Considering the media typification of the 462 single news items as paid media, spontaneous media, or organic media [25] (Table 5), it is possible to verify that the volume of spontaneous news disseminated (n = 282) is nearly two times higher than that of organic media (n = 172, state-owned media run by MoH and SNP, added to the owned media run by autarchies, state and municipal health secretariats. This reflects the campaign’s effectiveness in sparking voluntary interest from the Brazilian business media, which has incorporated the syphilis issue into its agenda, thus strengthening the tendency of the press to transfer its interests to society’s agenda [13].

Figure 5 displays the agenda-setting effects between media (intermedia), media and organizations, and organizations (inter-organization). There were seven clusters from one national media to another (clusters 14, 16, 48, 53, 54, 56, and 59), whereas cluster 16 starts in a public organization and then sets the media. Additionally, four intermedia agenda-setting effects were wielded among regional news media coverage (clusters 24, 32, 43, 50). Figure 5 also evidences one case of agenda-setting effect between a national media organization and a private communication organization (cluster 28) and another between a national media and a public organization (cluster 52). From the regional media to a public organization, one agenda-setting process was identified (cluster 46). The cluster "0" makes the interesting movement of agenda-setting from one organization to another (inter-organization) that leads to publication in national media. And two other agenda-setting were from public organizations directly to the national media (clusters 2, 45). Seven agenda-setting were carried out from public organizations to regional media (clusters 4, 20, 21, 31, 36, 40, 44). Lastly, the agenda-setting between organizations running news pages was observed in five clusters (30, 33, 34, 37, 51). 

Thus far, there are 29 clusters wherein agenda-setting effects include spontaneous news, i.e., produced without the action or agency of the organizations responsible for the communication campaigns of the SNP. The news reports published on webpages of organizations that wielded agenda-setting effects within other organizations (inter-organization) in an organic manner—meaning that such organizations are directly associated with the MoH or the SNP (state and municipal health secretariats)—comprised 13 clusters in total (clusters 8, 9, 12, 27, 35, 38, 39, 41, 42, 47, 55, 57, 58). In relation to the agenda-setting of the syphilis issue involving organizations, we also identified the “news recycling” phenomena in 12 clusters, where texts discussing the disease are reposted in different media spaces or updated by these organizations. Eight of these clusters were identified through spontaneous news (clusters 1, 5, 4, 6, 7, 10, 11, 19), and the other three were in organizations linked to the project or the MoH in an organic way (clusters 3, 13, 22). One of them was a result of paid media whose news was reutilized in the website of the institution responsible for the SNP, that is, the Laboratory for Technological Innovation in Health (LAIS) at the Federal University of Rio Grande do Norte (UFRN).

It is worth noting that, in cluster 14, the intermedia agenda-setting effects occur between journalistic companies within the same group. In Cluster 53, we identified a news report published on 05 November 2019 by the newspaper of national reach “Folha de São Paulo”. Such a piece was replicated on the website of the Regional Council of Medicine of Pernambuco on 06 November 2019 and by the newspaper “O Tempo”, from Belo Horizonte, Minas Gerais, on 07 November 2019. The latter case demonstrates such effects [12,13], whereby a news story conveyed in one of Brazil’s top newspapers was replicated two days later in a regional newspaper in the state of Minas Gerais.

Another significant piece of data we spotted in the clusters characterized by the intermedia agenda-setting process shows that all the 11 clustered processes occurred in 2018 (cluster 14: national media) and 2019 (24, 32, 43, 50: regional media; 16, 48, 53, 54, 56, 59: national media), years during which the SNP developed its communication campaigns.

A relevant fact can be observed in cluster 16, wherein the agenda-setting process stems from a public organization towards a national media and, subsequently, is transformed into an intermedia agenda-setting effect. Regarding cluster 16, the news report from RADIS magazine was reproduced twice by FIOCRUZ, its organization, through its online media sites (i.e., Portal Fiocruz and Portal ENSP), and it was featured on BBC News 10 months later. Such a publication led two reporters to investigate the subject, find new sources, interview them, and produce original material using the same structural approach and angle. In journalistic jargon, this phenomenon is called a “hook”, the anchor of a news report. Shortly after that, besides publishing it on its website, BBC News also sold it to UOL and Globo group, which reproduced it on their portals, quoting the initial source in the credits.

Still, regarding the spontaneous news generated by the media, we identified the agenda-setting effect (i) in a news space of a private organization (cluster 28); (ii) from national media on a public organization (cluster 52), and (iii) from regional media on a public organization (cluster 46). Conversely, we perceived an inter-organizational agenda-setting effect between two public organizations that wielded an influence on national media (cluster 0).

In the case of cluster 0, the original news report explaining “what is syphilis”, published on 9 May 2016 on the Health Secretariat of São Paulo website, was replicated eight times. The first was 16 months later, on 21 October 2017, when the Health Secretariat of Bahia and the Health Surveillance Department of Santa Catarina updated the news report. The year following, on 24 February 2019, this department updated the publication content. Then, on 29 April 2019, it was the turn of the news website of the Federal University of Paraíba (UFPB) to add information about SIP and CS. About four months later, on Aug. 16, 2019, the MoH website recaps the same information in a new section, “Health from A to Z”, with a focus close to the UFPB website, highlighting the care of SIP and CS infections. After two months, during the period dedicated to the SNP campaign’s actions, the text was updated thrice: on October 10, October 21, and 27 October 2019, on the news website of the MoH Virtual Library, the “Catraca Livre”, and MoH’s official website. This was a news report that provides all the epidemiological features of syphilis. Therefore, a news report with organic characteristics generated by the governmental agenda, which disseminated through news spaces of governmental instances focused on health. Interestingly, it was also published on an alternative news website, namely Catraca Livre, characterizing the publication as spontaneous media. The spontaneous media, i.e., the broadcasting of news about syphilis without the onus or initiative of those responsible for the public communication campaign, is indicative that the topic has achieved relevance in the media agenda and influenced the public’s agenda.

Moreover, news stories that resulted from the agenda-setting influence of a public organization on national media (clusters 2 and 45) can be considered spontaneous media, as can be the case when agenda-setting effects by public organizations were echoed in regional media (clusters 20, 21, 31, 36, 40, 44, 49). The inter-organizational agenda-setting effect between organizations (health, societal, or educational institutions) that disseminate the syphilis topic on their portals and websites was present as spontaneous news in five clusters (30, 33, 34, 37, 51). The recycling of spontaneous news was mapped out in eight clusters (1, 5, 4, 6, 7, 10, 11, 19).

In an organic format, thirteen clusters were characterized considering the news stories that exerted the inter-organizational agenda-setting effect among organizations directly related to the MoH, State, and Municipal Health Secretariats (clusters 8, 9, 12, 27, 35, 38, 39, 41, 42, 47, 55, 57, 58). Another three clusters resulted from recycled organic news (clusters 3, 13, and 22). Additionally, the agenda-setting arising from the republication of the news posted on the LAIS/UFRN website, paid for by the 2019 communication campaign of the SNP, rendered a recycled news item aired in the Metrópoles newspaper (Cluster 17).

Of note, the Google search engine returned, on different dates, identical URLs that were grouped into five clusters (15,18, 23, 26, 29). To some extent, this indicates that the news repeatedly appeared at different moments, thus contributing to sustaining the topic’s prominence. Further, cluster 25 presented two web pages containing a table with the historical series of SIP and CS in the city of São Paulo, pages that experts incorrectly categorized. On the other hand, given the number of news reports evaluated (from 1049 to 619), this discrepancy does not significantly differ from the results.

The effects of agenda-setting on the media, for instance, regarding issues like syphilis and the possible transfer of salience to the public’s agenda, is not immediate and take place in the medium-term [12]. Thus, there is a “latency” period between the stimulus (appearance of a topic on the media agenda) and the repercussion/response on the public’s agenda (relevance transfer process) [13]. This response or repercussion on the public’s agenda can be measured, for example, through comments on social media, the number of searches for specific keywords (e.g., syphilis) on digital search engines, and academic research initiatives that may have been motivated by reading published news about the disease.

Based on the “intermedia agenda-setting” theory [13], this study predicts and provides an approximate temporal range related to the communication strategy of an organized and structured public health campaign. It provides a mapping that can be used as a model and transfer element for planning future communication processes in public health crises. Simultaneously, it reveals the importance and power of journalistic works, which provide a source and can be replicated by other media outlets.

Therefore, the present work poses new research questions from a computational and communication perspective. Questions such as “what are the narrative elements themselves?” and “what formats can enhance and replicate communicative events in a way that can produce solid clusters on the public media agenda?” are highly relevant questions for future health campaign blueprints.

Pinto et al. [2] present a multidimensional exploratory analysis of the syphilis epidemic in Brazil. These authors analyze the growing interest of the Brazilian population in the subject on search engines and increased media coverage for the respective period. They also observed the positive impact of the SNP, reporting the rise in population syphilis test uptake and the drop in cases, reversing the growth trend of syphilis after nearly a decade.

Moreover, in [26], the same group of researchers performed a systematic literature review on approaches for evaluating public health campaigns. They concluded that such campaigns should be assessed from multiple perspectives to grasp the overall reach of mass communication. In addition, they observed a shortcoming in the impact evaluation of public health campaigns in terms of online data usage (online news and other user-generated online content). Further, these scholars highlighted some overlooked topics in the existing literature, e.g., the analysis of publications and research projects on the subject.

Hence, the results of the syphilis online news clusters for three periods of the SNP campaign suggest it encouraged the transfer of salience of the syphilis issue from the media to the public’s agenda, with an increase in the replication of news items. In conclusion, the sum of the during-campaign and post-campaign periods reflects the success of the agenda-setting process as a time of sustained media coverage, with an increase in replications, in addition to the consolidation of the intermedia agenda-setting, with an 87.3% correspondence of replications over time. It is noteworthy that, given the epidemic scenario of syphilis in Brazil [27], developing a sustainable mass communication agenda in health and demonstrating evidence of its sustainability is relevant for conducting public health policy, especially in crisis scenarios.

## 5. Conclusions

The present multidisciplinary research analyses public health communication through computer science approaches. A vast body of literature on news clustering [17,18,19] focuses on evaluating the performance of the tools developed. Hence, scholars had a set of previously classified data and executed an algorithm to verify the accuracy of the matches or algorithm performance. By contrast, this study built a tool not merely limited to method analysis, as it extrapolates the results and aids in understanding a public health campaign.

Using a computational approach to evaluate the effectiveness of communication regarding a public health problem, the results presented in this paper demonstrate that the volume of news about the syphilis issue has been on the rise in recent years and gained popularity after the launch of the “Syphilis No!” project. Likewise, its popularity was boosted during the SNP campaign and kept increasing even after it ended.

In the context of global health, there is a process of digital transformation of health [28], especially after the severe scenario caused by the COVID-19 pandemic [29]. Therefore, using computational methods and incorporating technologies into health systems that can improve and qualify decision-making is essential, especially in scenarios of health crises, where there is a need to rationalize efforts due to the scarcity of public resources and inputs [30].

Brazil is a continental country with significant social and cultural differences. Therefore, epidemics such as syphilis, whose problem is the multifactorial order, become even greater challenges to be faced. The results discussions presented in this article highlighted an essential public health dimension: communication. Public health crises, such as syphilis in Brazil and COVID-19, demand different actions in different dimensions to produce more effective responses. The effectiveness of the response is also a resilience inducer [31,32,33] and responsiveness of the health system in the face of health emergencies. The pandemic has provided several lessons in this regard.

The results of this study demonstrate, mainly, that mass communication in public health, when planned and executed based on scientific evidence, constitutes an essential tool for conducting policy. This is because the qualified use of this tool can contribute to the process of inducing resilience in the health system in the face of a health crisis. Therefore, countries such as Brazil, with a continental dimension, high population, inequalities, and a complex health system, require decision-makers to use technological tools [34,35,36] that can better qualify the massive health communication campaigns development, mainly when financial resources are scarce, and the results need to be more effective [37,38,39,40].

Thus, a communication action in public health is more efficient when it reaches repercussions in informative spaces of the media and organizations of society, and, mainly, when it manages to expand its repercussion by reverberating in other regional and national information organizations and media, expanding its power of reach for different audiences by the different means of social communication. An aspect that favors the capillarization of information, health education, and health promotion in coping with public health crises. In this article, in addition to demonstrating the similarity analysis, it was possible to observe the communication strategy efficiency in public health adopted in the “Syphilis No!” Project.

## Figures and Tables

**Figure 1 ijerph-19-17049-f001:**
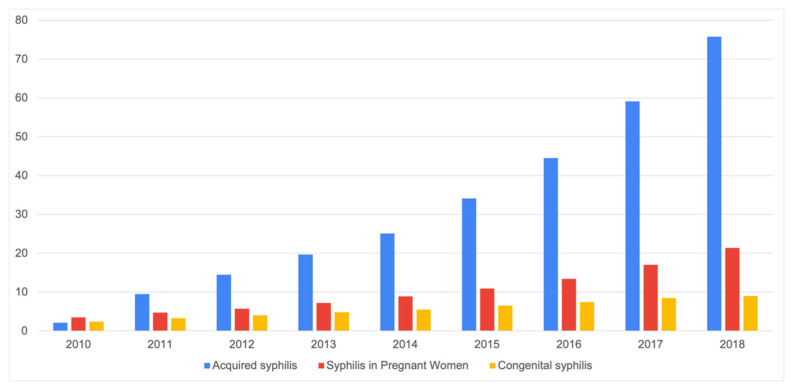
Reported cases of syphilis in Brazil between 2010–2018. Source: Ministry of Health (MoH).

**Figure 2 ijerph-19-17049-f002:**
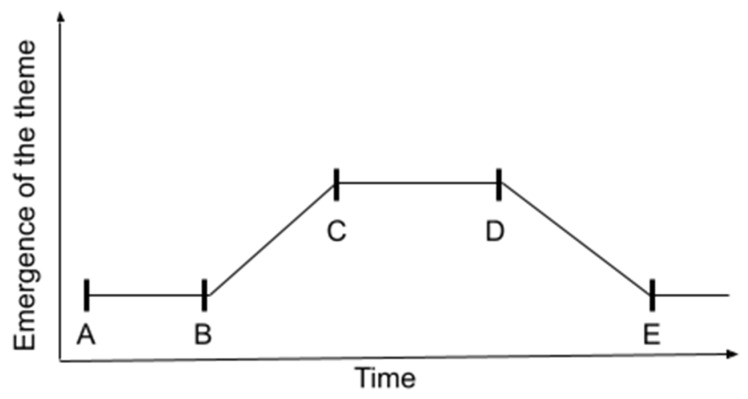
Timeframe for agenda-setting effects. Adapted from [13].

**Figure 3 ijerph-19-17049-f003:**
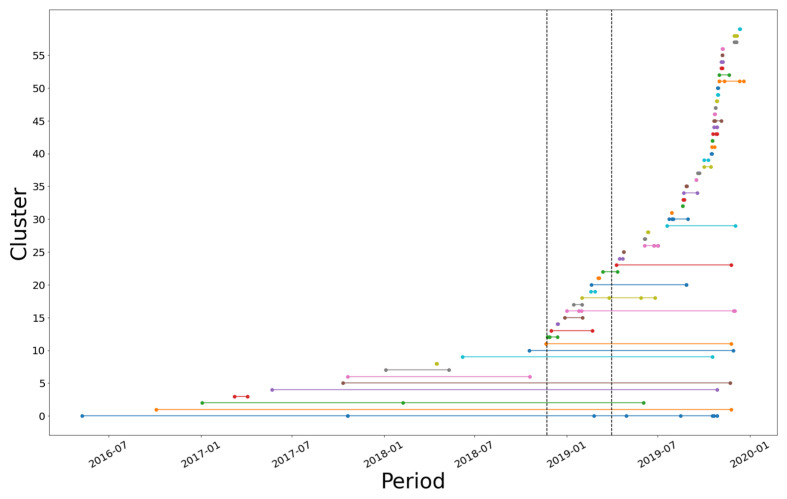
Replication of news stories in the period analyzed, grouped into clusters of at least two news items.

**Figure 4 ijerph-19-17049-f004:**
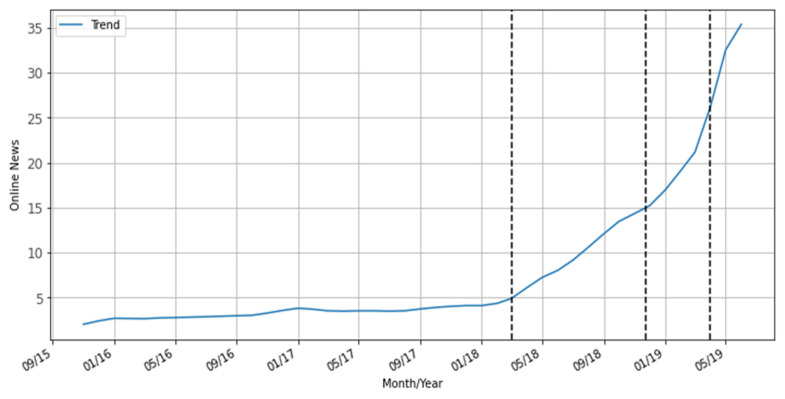
Trend graph of online news stories on syphilis in Brazil between 2015–2019.

**Figure 5 ijerph-19-17049-f005:**
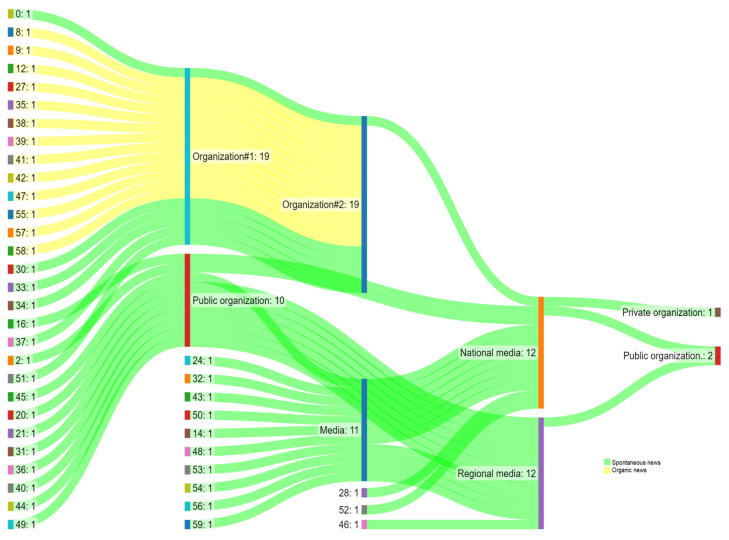
Scheduling processes between media (intermedia), between media and organizations, and between organizations (inter-organization).

**Table 1 ijerph-19-17049-t001:** Clusters and their respective sizes.

Cluster	Number	Cluster	Number	Cluster	Number
0	9	20	3	40	2
1	2	21	2	41	2
2	3	22	2	42	2
3	2	23	2	43	3
4	2	24	2	44	3
5	2	25	2	45	3
6	2	26	2	46	2
7	2	27	6	47	2
8	2	28	2	48	2
9	2	29	2	49	2
10	2	30	5	50	2
11	2	31	2	51	5
12	3	32	2	52	2
13	2	33	2	53	3
14	2	34	2	54	3
15	2	35	3	55	2
16	6	36	2	56	2
17	2	37	2	57	6
18	4	38	2	58	3
19	2	39	2	59	2

**Table 2 ijerph-19-17049-t002:** Number of news reports and clusters identified between 2015–2019.

Time Period	News Reports		Clusters	
	Number	%	Number	%
1 January 2015–21 November 2018	164	26.49%	18	11.46%
22 November 2018–31 March 2019	85	13.73%	24	15.29%
1 April 2019–31 December 2019	370	59.77%	115	73.25%
Total	619	100%	157	100%

**Table 3 ijerph-19-17049-t003:** Source of news publication.

Source	Amount
National Media	68 (14.72%)
Regional Media	86 (18.61%)
Ministry of Health, State and Municipal Health Secretaries	162 (35.06%)
Health or Societal organizations or Educational Institutions	146 (31.60%)
Total	462 (100%)

**Table 4 ijerph-19-17049-t004:** Journalistic Genre.

Category	Number
News	295 (63.85%)
Opinion Articles	73 (15.80%)
News Reports	71 (15.37%)
Other	23 (4.98%)
Total	462 (100%)

**Table 5 ijerph-19-17049-t005:** Types of Media.

Category	Number
Paid Media	8 (1.73%)
Spontaneous Media	282 (61.04%)
Organic Media	172 (37.23%)
Total	462 (100%)

## Data Availability

The data sets used and analyzed during the current study are available from the corresponding author on reasonable request.

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
