# Peer review of "Similarity Analysis in Understanding Online News in Response to Public Health Crisis"

_ijerph, 2022, doi:10.3390/ijerph192417049_

Round 1
Reviewer 1 Report
This paper aim to explore the similarity of syphilis online news to comprehend how public health communication interventions influence media coverage of the syphilis issue. Whether "Syphilis No!" campaign have effectively prompted the “intermedia agenda-setting” effect or not. However,theirs findings presented in this study show that the volume of syphilis-related news reports has increased in recent years and gained popularity after the SNP started, this is only a descriptive results, more deeper understanding should be analyzed.
1. The index that quantified the effectiveness of communication actions should given more explanation.
2. Figure 5 displays the intermedia agenda-setting effects between different media and organizations and within organizations, more media specific suggestions should be given.
3. In the introduction and discussion, although the content is very long, the core ideas are diluted and not highlighted because of their length. It is suggested to revise them.
- 4. The limitation of this study should be given.
Reviewer 2 Report
We consider that this proposal needs to delve into all the epigraphs, steps, of an academic process; that is to say, to strengthen the state of the art review, to explain and justify the objectives and hypothesis, to deepen into the methodology employed and explain why it is suitable, to add both epigraphs considering the 'discussion' and 'further research' implications and, as a result of all the previous ones, to extend the conclusions, so brief right now.
Reviewer 3 Report
Thank you for the opportunity to review the article “Similarity Analysis in Understanding Online News in Response to Public Health Crises”. The paper addresses an interesting and well researched theme in the recent period about the ways online news and digital information determine or influence how public health communication interventions generate media coverage of the specific issues. The paper is also in line with the section “Health Communication and Informatics” and with the special issue was submitted to: “Communication and Information Technology in Healthcare Management”.
This study represents a solid effort in the field approached. It is constructed in a mature manner, following the publication standards of the journal, discussing the subject that needs to be comprehensively analyzed because “governmental sectors have been increasingly investing in public health campaigns to promote public awareness regarding disease risk, as well as to disseminate diagnostic tests and treatment methods”, as the authors underline.
Also, the study is written in an adequate manner, with a specific review of the literature and robust research design. The results are presented clearly and coherently, using visuals and text. The tables presented in the paper were relevant to explore the results of the research and the ways these were adapted to the explanations in the text.
As the authors say, one conclusion of this paper assumes that “the sum of the during-campaign and post-campaign periods reflects the success of the agenda-setting process as a time of sustained media coverage”.
Moreover, there are some point-by-point observations that should be addressed in this revision.
- Try to use better quality images for Figure 1 and bigger characters for Figure 3. As much as the reader is interested in seeing the details of the figure, the possibility of doing that is very limited.
- The “similarity” concept from the title is not presented in the paper linked to the methodology of online news clustering based on similarity analysis. This term should be detailed more, maybe in the research objectives, too.
- In the Conclusions section, please add the novelty factor of this study.
- In the Reference section, some of the co-authors of this study are overciting their own works. This situation would not be a problem if the overall number of the consulted literature would be higher.
Round 2
Reviewer 1 Report
-
I agree to publish it in its present form。